# Packaged Foods Labeled as Organic Have a More Healthful Profile Than Their Conventional Counterparts, According to Analysis of Products Sold in the U.S. in 2019–2020

**DOI:** 10.3390/nu13093020

**Published:** 2021-08-29

**Authors:** Aurora Dawn Meadows, Sydney A. Swanson, Thomas M. Galligan, Olga V. Naidenko, Nathaniel O’Connell, Sean Perrone-Gray, Nneka S. Leiba

**Affiliations:** 1Environmental Working Group, Washington, DC 20009, USA; sydney@ewg.org (S.A.S.); thomas.galligan@ewg.org (T.M.G.); olga@ewg.org (O.V.N.); 2Department of Biostatistics and Data Science, Division of Public Health Sciences, Wake Forest School of Medicine, Wake Forest University, Winston-Salem, NC 27109, USA; n.oconnell@wakehealth.edu; 3Novi Connect, Inc., Larkspur, CA 94939, USA; sean.perronegray@gmail.com; 4Amazon.com, Inc., Seattle, WA 98109, USA; nsleiba@gmail.com

**Keywords:** organic products, conventional products, product attributes, ultra-processed foods, packaged foods, processed foods, hyper-palatable foods, nutritional quality, nutrient composition, cosmetic additives

## Abstract

The organic food market’s recent rapid global growth reflects the public’s interest in buying certified organic foods, including packaged products. Our analysis shows that packaged foods containing fewer ingredients associated with negative public health outcomes are more likely to be labeled organic. Previous studies comparing organic and conventional foods focused primarily on nutrient composition. We expanded this research by additionally examining ingredient characteristics, including processing and functional use. Our dataset included nutrition and ingredient data for 8240 organic and 72,205 conventional food products sold in the U.S. from 2019 to 2020. Compared to conventional foods, organic foods in this dataset had lower total sugar, added sugar, saturated fat and sodium content. Using a mixed effects logistic regression, we found that likelihood of classification as organic increased as sodium content, added sugar content and the number of ultra-processed ingredients and cosmetic additives on the product label decreased. Products containing no trans-fat ingredients were more likely to be labeled organic. A product was more likely to be classified “organic” the more potassium it contained. These features of organic foods sold in the U.S. are significant because lower dietary ingestion of ultra-processed foods, added sugar, sodium and trans-fats is associated with improved public health outcomes.

## 1. Introduction

People who purchase organic foods often cite “health” as their reason for choosing organic over conventional foods [1,2,3,4,5,6,7,8] and consider organic processed foods as more healthful than conventional processed foods [9,10]. In order for a food to be U.S. Department of Agriculture (USDA) certified organic, it must comply with U.S. federal regulations regarding cultivation, formulation and production [11]. These regulations include limiting the use of synthetic pesticides, fertilizers, antibiotics and food additives; fewer than 40 synthetic additives are permitted in processed organic products [11,12]. As the availability of organic packaged and processed foods continues to grow [13], both shoppers and researchers raise questions about comparative healthfulness of conventional and organic packaged and processed foods. A cross-sectional study of the U.S. packaged food and beverage supply in 2018 reported that 71 percent of the products were classified as ultra-processed [14]. Approximately 60 percent of the calories consumed by U.S. persons over the age of two years old came from ultra-processed foods and beverages, according to two separate studies spanning 2000–2012 [15,16]. U.S. youths aged 2–19 years exhibit even higher consumption of calories from ultra-processed foods, averaging 67 percent—a 6 percent increase in intake over the last two decades [17]. Data on the Canadian diet, which contains a similar proportion of calories from ultra-processed foods, shows a 50 percent increase in consumption of ultra-processed foods from 1953 to 2001 [18].

For many varieties of processed foods, both conventional and USDA-certified organic options are now available. According to a recent systematic review, a diet high in organic foods has been epidemiologically linked to lower risks of some cancers, overweight/obesity and type 2 diabetes [19]. There are several possible mechanisms linking organic foods to health benefits, including reduced exposure to synthetic pesticides, differences in lifestyle factors between individuals who purchase organic and conventional foods, and differences in nutrient profiles between organic and conventional foods [19]. One additional possible mechanism, relevant to the packaged food market, is potential differences in the extent of processing between organic and conventional food products.

Two prior studies of the healthfulness of organic packaged or processed foods focused exclusively on nutrients and nutrient profiles. A 2014 study conducted in the United States used a nutrient profiling system, NuVal [20], to compare 829 organic and conventional ready-to-eat breakfast cereals [21]. A 2020 study conducted in Italy reviewed the differences in calories, total fat, total carbohydrates, total sugar, total protein, saturated fat and salt content in 569 food product pairs [22]. Both studies concluded that organic packaged foods were not of superior nutritional quality to conventional foods according to nutritional parameters they considered. However, these studies had small sample sizes and limited coverage of the processed food market, with one study only including one category, ready-to-eat breakfast cereals. In this study, we used a dataset representative of the U.S. packaged and processed food market to overcome the limitations in prior studies, including both nutrient information and degree of processing, as proposed by Poti et al. (2017), to assess healthfulness of food products.

Degree of processing has been proposed as a useful metric for assessing processed food healthfulness [23]. Packaged and processed foods can be categorized along a continuum from unprocessed to ultra-processed based on the nature, extent and purposes of the industrial processes used to produce them [24]. Two studies that evaluated the degree of food processing of national marketplaces (one in New Zealand and the other in the United Kingdom) used nutrient profiling methods to show that the nutritional quality of a diet/food decreases along the processing continuum [25,26]. Generally, the higher the degree of processing of foods, the more calorically dense [27,28,29,30,31] and higher in trans-fat [31,32,33,34], saturated fat [15,29,30,31,32], added sugars [31,32,35,36,37] and sodium [15,26,34,35] they are, and thus, highly processed foods are regarded as generally of poorer nutritional quality [23,25,29]. These components are linked to adverse health outcomes: trans-fat intake is linked to increased risk for type 2 diabetes, cardiovascular disease and risk of mortality [38,39]; sodium intake is also linked to increased risk of mortality [40] and high blood pressure [41]; added sugar intake is associated with a higher risk of cardiovascular disease mortality [42]; while low saturated fat intake is associated with a reduced risk of cardiovascular disease [43]. The U.S. Dietary Guidelines state that “the food components of added sugars, solid fats and sodium…are consumed in excess of recommendations,” highlighting these as components to limit [44].

Higher intake of ultra-processed foods is associated with a concomitant decrease in disease-preventing vegetable intake [26,45] and increased risk of wheezing or asthma in teens [46,47], depression [48,49], mortality [50,51,52,53] and chronic diet-related diseases such as metabolic syndrome [54,55,56,57], obesity [23,58,59,60,61,62], type 2 diabetes [63,64], hypertension [65], heart disease [66], stroke [66] and cancer [67,68,69] in adults. A meta-analysis of 43 studies on ultra-processed foods and chronic disease risk found consumption of ultra-processed food was associated with an increased risk of overweight, obesity, abdominal obesity, all-cause mortality, metabolic syndrome and depression in adults [47]. For these reasons, Canada, Mexico and Brazil, among others, recommend that people limit consumption of ultra-processed foods [70,71,72,73,74].

In addition to nutrient components, ultra-processed foods also contain a variety of food additives, some of which are known to be harmful. Evidence suggests that consumption of food additives found in ultra-processed foods, including additives added to affect the perception of sensory characteristics (such as artificial colors, flavors and sweeteners, hereafter referred to as cosmetic additives), as well as nitrates, phosphates and preservatives, may increase chronic disease burden [67,75,76,77,78,79,80]. In a 2018 policy statement, the American Academy of Pediatrics raised concerns about children’s exposure to food additives (specifically, “colorings, flavorings and chemicals deliberately added to food during processing”) and adverse health effects in children [81]. “Cosmetic additives” are defined as ingredients used to artificially enhance the food’s sensory characteristics (i.e., produce desirable characteristics or mask undesirable characteristics) and increase their sensory appeal and palatability (e.g., flavors, flavor enhancers, colors and sweeteners) [82]. Hyper-palatable foods are noted for their ability to trigger/alter neurobiological reward systems and use of additives to maximize consumption of such foods, thus contributing to high public health costs [83,84,85]. Schulte et al. (2015) found that a food’s degree of processing was a strong positive predictor of its potential to be associated with addictive-like eating behavior [86]. Hyper-palatable foods are typically classified based on nutritional characteristics such as high caloric, sugar, fat and/or sodium content [87]. Cosmetic additives enhance the hyper-palatability of the food, for example by increasing their visual appeal [88]. Following published research, we use cosmetic additive content as an indicator of hyper-palatable foods that may promote obesity and chronic disease [67,75,76,77,78,79,80,81].

The aim of this study is to determine whether there are differences in the healthfulness of organic and conventional packaged and processed foods by comparing degree of processing (as indicated by ultra-processed ingredient content), hyper-palatability (as indicated by cosmetic additive content) and nutritional quality across a large dataset representative of the U.S. packaged and processed food market.

## 2. Materials and Methods

### 2.1. Food Product Dataset

The packaged and processed food dataset used for this study was purchased from Label Insight, A Nielsen IQ Company, which captures images of product packaging and compiles and digitizes label information for food products sold in U.S. supermarkets and big box stores. Label Insight’s product data, gathered from national, regional, specialty and local grocery retailers, represents 85% of total annual sales volume for consumer products sold in the United States [30]. The dataset included nutrition and ingredient data for 72,205 conventional and 8240 organic food products captured by Label Insight between the dates of 31 December 2018 and 7 January 2021. Ingredient parsing errors were reviewed and resolved in the overall dataset.

### 2.2. Ingredient Classification

The U.S. Department of Agriculture defines “unprocessed agricultural products” as agricultural products that have not been subject to any processing beyond heating, refrigerating and freezing, peeling, cutting, grinding, drying or pasteurization [89]. Similarly, the International Food Information Council’s definition of processed food categories is broad, not distinguishing between categories and leading to potential overlap [90]; it was also found to underestimate the percentages of foods that are highly processed [91]. Here we distinguish products along the continuum of processed foods, ranging from unprocessed to ultra-processed by applying the NOVA framework, which classifies foods and drinks into one of four categories: Group 1, Unprocessed or minimally processed foods; Group 2, Processed culinary ingredients; Group 3, Processed foods; Group 4, Ultra-processed food and drink products, as described in the published literature [24,82,92,93,94,95,96]. In addition to products, ingredients can also be classified according to the degree of processing [24]. The presence of just one ultra-processed ingredient can be sufficient to classify a product as ultra-processed [24,93]. We classified all ingredients in our dataset based on the NOVA system and determined how many ingredients in each product were ultra-processed. Cosmetic additives, a subset of ultra-processed ingredients, were defined by Monteiro et al. and include flavors, flavor enhancers, colors, emulsifiers/emulsifying salts, sweeteners, thickeners, anti-foaming agents, bulking agents, carbonating agents, foaming agents, gelling agents and glazing agents [97].

To determine which ingredients are ultra-processed, and which of those are cosmetic additives, we reviewed the function of every ingredient in our product dataset by referencing the U.S. Code of Federal Regulations [98], Evaluations of the Joint Food and Agriculture Organization of the United Nations/World Health Organization Expert Committee on Food Additives [99], and the Codex General Standard for Food Additives database [100]. Ultra-processed ingredients fall into two categories: (1) classes of food additives with an ultra-processing function and (2) food substances of no or rare culinary use that are used only in the manufacture of ultra-processed foods [82,93]. Food substances of no or rare culinary use (e.g., pea protein isolate) were classified based on similarity to a directly specified ultra-processed substance (e.g., soy protein isolate) [93]. In cases where the regulatory references listed multiple functions for an ingredient, we determined the primary function by reviewing the ingredient lists for a randomized sample of products within each of the product categories in which the ingredient was used and determined the function in those products. When multiple functions were applicable for an ingredient, we assigned the functions according to product “aisle”, a broad categorization for categories of related products. Appendix A provides a summary list of the ingredients we identified as ultra-processed and as cosmetic additives.

### 2.3. Overall Dataset Analysis

The primary goal of this study was to assess the relationships between a suite of product characteristics and USDA-certified organic status. Our primary statistical outcome was “organic status,” and our set of initial predictors included total ingredient count, ultra-processed ingredient count, cosmetic additive count, non-cosmetic additive count, calories, total sugar, added sugar, saturated fat, sodium, potassium and trans-fat. From these variables, we calculated for each product the percent of ultra-processed ingredients that were cosmetic additives and the percent of total sugar that was added sugar. Calories, total sugar, added sugar, saturated fat, sodium and potassium were each evaluated and reported in terms of mass per 100 g serving. Trans-fat is a binary variable for whether trans-fat-containing ingredients were listed in the ingredient list. Ingredients classified in this study as containing trans-fat are listed in Appendix A. Summary statistics were calculated for each variable of interest stratified by organic status, with continuous variables reported using mean, median and standard deviation, and categorical variables reported using frequency and percentages.

### 2.4. Imputation of Missing Data and Sensitivity Analyses

Not all nutrients appear on all food labels; added sugar, potassium and saturated fat were the most omitted nutrients on nutrient labels in our dataset (Table 1). Therefore, we used imputation by Random Forest [101,102] to impute missing data for our primary analysis. Due to its robustness for addressing multicollinearity issues, variables that were highly correlated, such as grams of total sugar and grams of added sugar, were both included in the imputation model, in addition to food “aisle” (a broad categorization variable consisting of 51 groups), total ingredient count, ultra-processed ingredient count, cosmetic additive count, non-cosmetic additive count, calories, saturated fat, sodium, potassium and trans-fat. The variables “percent of sugar that was added” and “percent of ultra-processed ingredients that were cosmetic additives” were calculated following the imputation of missing values for sugar, added sugar, ultra-processed ingredient count and cosmetic additive count. We conducted our primary analysis on the imputed data set. We further conducted two sets of sensitivity analysis: (1) complete case analysis (i.e., an analysis including only those products with values for all included variables, corresponding to 3661 organic and 28,596 conventional products), and (2) assuming missing values to be 0 (e.g., missing data for added sugar is assumed to be 0 g of added sugar).

### 2.5. Modeling Statistical Relationships between Organic Certification and Product Characteristics

The log-odds of a food product being USDA-certified organic were modeled using a generalized linear mixed model with logit link function, nested within food category (a narrow categorization variable nested within aisle, *n* = 996; Appendix A) via random intercepts. This approach allows a consideration of the collective effects of nutrient and ingredient characteristics on organic status in a comprehensive manner while accounting for food category, a likely strong driver of differences in these variables among foods. The independent variables included in the model were: saturated fat (10 g/100 g serving), added sugar (10 g/100 g serving), sodium (100 mg/100 g serving), potassium (100 mg/100 g serving) and trans-fat (binary Yes/No) and either ultra-processed ingredient count or cosmetic additives ingredient count. Calorie count was excluded given its dependence on the combination of added sugar and saturated fat. Data were analyzed using R software version 4.0.3, and SAS software version 9.4, and statistical significance was determined by a *p*-value ≤ 0.05.

## 3. Results

### 3.1. Data Overview

Table 2 provides the summary statistics for the presence of ultra-processed ingredients and cosmetic additives with no stratification by or correction for food aisle or category. Mean numbers of ultra-processed and cosmetic additives were clearly different between organic and conventional products in the overall dataset.

Table A1 lists the mean concentrations of nutrients in organic and conventional food products in our dataset, with no stratification by or correction for food aisle or category. We include means for both for the subsets of products that have nutrient values listed on the label, and the overall product dataset whereby missing nutrient values were imputed, as described in Section 2. In Table A2 we list the mean concentrations of nutrients, and minimum and maximum values for organic and conventional food products in our dataset stratified by 51 aisles for products that had complete nutritional information available.

Compared to conventional foods, organic foods in this dataset had lower total sugar, added sugar, saturated fat and sodium content (Table A1). Further, 685 (8%) of 8240 organic products contained an ingredient that we classified in this study as trans-fat containing (Appendix A), while 40% of conventional products, or 28,841 products out of 72,205, contained such ingredients. We note that since the mean values listed in Table A1 apply to the entire dataset, without separation of product categories, mean concentrations of nutrients per 100 g serving presents a very generalized characterization of products in the marketplace. Potentially erroneous nutrient values due to labeling errors or rounded values may impact the accuracy and precision of calculated values listed here. Analysis of all products in the dataset, including products with imputed missing data produced results that were overall similar to the mean concentrations calculated for the subsets of products with complete nutrient information (Table A1).

### 3.2. Modeling Relationships between Organic Status and Nutrient and Ingredient Characteristics

Table 3 and Table 4 present the results from our primary mixed logistic regression modeling in terms of odds ratios and 95% confidence intervals for the entire product dataset. Products with missing nutrient values were assigned imputed values, as detailed in Section 2.4. The independent variables included in the model were: saturated fat (10 g/100 g serving), added sugar (10 g/100 g serving), sodium (100 mg/100 g serving), potassium (100 mg/100 g serving) and trans-fat (binary Yes/No) and either ultra-processed ingredient count or cosmetic additives ingredient count. The mixed effects model, with food category included as a random effect, allowed us to consider the collective effects of nutrient and ingredient characteristics on organic status in a comprehensive manner while accounting for food category, a likely strong driver of differences in these variables among foods. We found that the odds of a product being certified organic decreased by 32% for each additional ultra-processed ingredient it contained (Odds Ratio 95% CI: 0.67–0.69, *p* < 0.001) (Table 3). The odds of organic classification decreased by 37% for each additional cosmetic additive ingredient a product contained (Odds Ratio 95% CI: 0.61-0.64, *p* < 0.001) (Table 4). Additionally, we found that the odds of organic status classification decreased with increasing added sugar and sodium and increased with increasing potassium (Table 3). The odds of organic classification decreased relative to the presence of ingredients classified in our study as trans-fat containing (Table 3, Odds Ratio 0.31, 95% CI: 0.28–0.34). Amount of saturated fat per serving was not significantly related to organic status (Table 3 and Table 4).

### 3.3. Sensitivity Analyses

Our complete case analysis assessed 3661 organic and 28,596 conventional food products for which all nutritional information were available for the parameters included in our mixed logistic regression model (Table 5 and Table 6). In the complete case analyses, we found similar directional and significant associations to the primary analysis of dataset with the imputed values. Specifically, for the complete case analysis (Table 5 and Table 6) we found decreased odds of organic classification with either increasing number of ultra-processed ingredients or increasing number of cosmetic additives; increasing added sugar and sodium content; and decreasing potassium, comparable to the models including the full dataset with imputed missing values (Table 3 and Table 4). For the complete case analysis, we noted a key difference regarding saturated fat. For the complete case dataset, the odds of being labeled organic decreased as saturated fat content increased, whereas there was no significant relationship between saturated fat and organic status in our assessment of the dataset with imputed missing values.

We also conducted a sensitivity analysis that assigned a value of zero to all nutrient parameter values used in our model that were missing from product labels in our dataset (Table A3 and Table A4). Assuming all missing values represent 0 (i.e., if added sugar was missing, we assume 0 g of added sugar), we find similar associations to the primary and complete case analyses for ultra-processed ingredients, cosmetic additives, sodium and potassium. For the dataset with missing nutrient parameter values assigned to zeros, no statistically significant effects were observed for added sugar and saturated fat (Table A3 and Table A4).

## 4. Discussion

To our knowledge, this is the first study to compare healthfulness of organic versus conventional packaged and processed foods comprehensive of the entire U.S. market. Further, this is the first study to use metrics of processing and hyper-palatability in addition to nutritional quality to assess healthfulness in packaged and processed foods by organic status. Earlier studies of organic and conventional packaged foods focused on limited number of products [21,22]. Our study, with a dataset representing 85% of packaged and processed foods sold in the United States, shows that organic packaged and processed foods in the U.S. are significantly different than conventional alternatives, based on metrics of nutritional quality as well as indicators of degree of processing and hyper-palatability.

Our analysis included 80,445 food products and beverages sold in the U.S. spanning 996 food categories. In our analysis of the U.S. packaged and processed food market, foods with fewer ultra-processed ingredients, fewer cosmetic additives, no ingredients containing trans-fat, less sodium, less added sugar and more potassium had greater odds of being labeled organic. Importantly, by including food category as a random effect within our analyses, we accounted for differences between food categories with respect to these variables, indicating that the observed differences are not limited to specific sub-categories of packaged and processed foods nor an artifact of the distribution of product samples across and within these categories. The findings for added sugar, sodium and potassium content were particularly interesting. Among the nutrient parameters included in our model, added sugar had the strongest effect, with, depending on the model, 12–15% decreased odds of a product being labeled organic for each 10 g increase in added sugar per 100 g product serving (Table 3 and Table 4). The effect size for sodium and potassium were smaller, yet statistically significant: 3–4% decreased odds of a product being labeled organic for each 100 mg increase in sodium per 100 g product serving; and 2% increased odds of a product being labeled organic for each 100 mg increase in potassium per 100 g product serving. Potassium content may indicate higher content of unprocessed/minimally processed ingredients, given that fruit, vegetables, vegetable juices, dairy, coffee and tea are the top five food sources of potassium in the U.S. diet [97] and that the use of potassium salts for potassium fortification in U.S. foods is currently uncommon (fewer than 5% of packaged and processed foods in our database include potassium salts). We did not include caloric density in our mixed logistic regression model due to high multicollinearity of calorie content with added sugar and saturated fat. However, the decreased odds of being classified organic with increasing added sugar (Table 3) and saturated fat (Table 5 and Table 6) may indicate a potential association between caloric density and organic status. Overall, we report several novel links between organic status and nutrients in packaged food sold in the U.S.

When limiting our analysis to only include products with complete nutritional label information (complete case analysis, Table 5 and Table 6), every 10 g increase in saturated fat (a nutrient parameter associated with diet-related chronic disease) per 100 g product serving was associated with a 15–18% (depending on the model) decreased odds of being labeled organic. Conversely, when assuming the missing values to be zero, as in our second sensitivity analysis, increasing saturated fat was not associated with organic status (Appendix B Table A3 and Table A4). Future work should further study the possible relationships between saturated fat and organic status in packaged and processed foods sold in the U.S.

Most prior assessments of differences in healthfulness between organic and conventional foods have focused on nutritional differences. Health experts have suggested that consumers, researchers and regulators need broader, more inclusive metrics to identify the most healthful foods and dietary patterns. Considering degree of processing in conjunction with nutritional quality may be one such approach [23], given the associations between ultra-processed foods and adverse health outcomes. Assessing hyper-palatability could also be useful as hyper-palatable foods disrupt appetite regulation, which can lead to compulsive overeating in part by stimulating “reward eating” and by suppressing satiety signaling (i.e., changing the homeostatic set point for energy balance/body weight) [103,104,105]. Here, with the finding that the odds of being labeled organic decreased as ultra-processed ingredient number or cosmetic additive number increased, we show that organic product certification can be a proxy for less ultra-processed and thus more healthful products.

The limitation of the present study is the difference between the number of organic and conventional products analyzed, which reflects the current state of the U.S. packaged food market. Further, while the organic packaged and processed foods analyzed here differed significantly from conventional packaged and processed foods, ultra-processed organic foods are not, in principle, as healthful as fresh, unprocessed or minimally processed organic foods. As such, shoppers should limit consumption of ultra-processed foods in general, in keeping with widely accepted dietary guidance. Additional research is necessary to confirm and characterize the relationships between organic processed or minimally processed—but not ultra-processed—packaged food consumption and health promotion.

## 5. Conclusions

Our study builds on prior research by using product metrics closely tied to adverse health outcomes such as proportion of ultra-processed ingredients in addition to nutritional quality to assess packaged and processed food healthfulness. We show that there are differences in the measures of healthfulness between conventional and organic packaged foods in the U.S. Overall, organic foods contain fewer ultra-processed ingredients and cosmetic additives and exhibit higher nutritional quality. These features of organic foods sold in the U.S. are significant because lower dietary ingestion of ultra-processed foods, added sugar, sodium and trans-fats is associated with improved public health outcomes.

## Figures and Tables

**Table 1 nutrients-13-03020-t001:** Frequency of missing nutrient values for specific nutrients, by percentage of the total of 8240 organic and 72,205 conventional products.

	Percentage of Organic Products with Missing Values	Percentage of Conventional Products with Missing Values
Calories	0.1%	0.1%
Total sugar	2%	3%
Added sugar	30%	29%
Saturated fat	16%	14%
Sodium	0.2%	0.4%
Potassium	42%	54%

**Table 2 nutrients-13-03020-t002:** Analysis of ingredient lists for 8,240 organic and 72,205 conventional products. All values are rounded to a whole digit.

	Organic		Conventional	
Ingredient Analyses	Arithmetic Mean (SD)	Median (Min, Max)	Arithmetic Mean (SD)	Median (Min, Max)
Ultra-processed ingredients (n)	2 (2)	1 (0, 18)	6 (6)	4 (0, 51)
Cosmetic additives (n)	1 (2)	0 (0, 11)	4 (4)	3 (0, 37)
Percent cosmetic additives per total Number of ultra-processed ingredients *	38 (43)	0 (0, 100)	55 (34)	62 (0, 100)

* For products containing zero ultra-processed ingredients and thus having an undefined value for percent cosmetic additives per total number of ultra-processed ingredients, a value of zero was imputed for this parameter.

**Table 3 nutrients-13-03020-t003:** Results from mixed logistic regression model of 8240 organic and 72,205 conventional products including the imputed missing nutrient data. This model included parameter “ultra-processed ingredients”.

	Organic Status		
Predictors	Odds Ratios	CI	*p*
Ultra-processed ingredients (n)	0.68	0.67–0.69	<0.001
Saturated fat (10 g/100 g)	0.94	0.87–1.03	0.176
Added sugar (10 g/100 g)	0.88	0.85–0.91	<0.001
Sodium (100 mg/100 g)	0.97	0.96–0.98	<0.001
Potassium (100 mg/100 g)	1.02	1.01–1.03	<0.001
Contains trans-fat: Yes	0.31	0.28–0.34	<0.001

**Table 4 nutrients-13-03020-t004:** Results from mixed logistic regression model of 8240 organic and 72,205 conventional products including the imputed missing data. This model included the parameter “cosmetic additives”.

	Organic Status		
Predictors	Odds Ratios	CI	*p*
Cosmetic additives (n)	0.63	0.61–0.64	<0.001
Saturated fat (10 g/100 g)	0.98	0.91–1.07	0.704
Added sugar (10 g/100 g)	0.85	0.82–0.88	<0.001
Sodium (100 mg/100 g)	0.96	0.95–0.97	<0.001
Potassium (100 mg/100 g)	1.02	1.01–1.03	<0.001
Contains trans-fat: Yes	0.27	0.24–0.30	<0.001

**Table 5 nutrients-13-03020-t005:** Results from mixed logistic regression model of 3661 organic and 28,596 conventional products that had complete nutritional information for parameters included in this model. This model included the parameter “ultra-processed ingredients”.

	Organic Status		
Predictors	Odds Ratios	CI	*p*
Ultra-processed ingredients (n)	0.70	0.69–0.72	<0.001
Saturated fat (10 g/100 g)	0.82	0.71–0.93	0.003
Added sugar (10 g/100 g)	0.95	0.89–1.00	0.065
Sodium (100 mg/100 g)	0.93	0.91–0.95	<0.001
Potassium (100 mg/100 g)	1.01	1.00–1.02	0.004
Contains trans-fat: Yes	0.31	0.27–0.35	<0.001

**Table 6 nutrients-13-03020-t006:** Results from mixed logistic regression model of 3661 organic and 28,596 conventional products that had complete nutritional information for parameters included in this model. This model included the parameter “cosmetic additives”.

	Organic Status		
Predictors	Odds Ratios	CI	*p*
Cosmetic additives (n)	0.64	0.62–0.66	<0.001
Saturated fat (10 g/100 g)	0.85	0.74–0.98	0.021
Added sugar (10 g/100 g)	0.92	0.87–0.98	0.008
Sodium (100 mg/100 g)	0.92	0.90–0.94	<0.001
Potassium (100 mg/100 g)	1.01	1.00–1.02	0.008
Contains trans-fat: Yes	0.28	0.24–0.32	<0.001

## Data Availability

Dataset used in this study was purchased from Label Insight, A NielsenIQ Company, labelinsight.com (accessed 26 August 2021). The dataset is viewable at.ewg.org/foodscores (accessed 26 August 2021).

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
