# Peer review of "Packaged Foods Labeled as Organic Have a More Healthful Profile Than Their Conventional Counterparts, According to Analysis of Products Sold in the U.S. in 2019–2020"

_nutrients, 2021, doi:10.3390/nu13093020_

Round 1

Reviewer 1 Report

The study appears well conducted. There are a few methodological issues that need addressing, noted below. Overall, I do feel the language is a bit forceful. The authors did not address if any of the foods meet independent criteria for “healthy options”, such as per the NPI.

The wording of the title is awkward. I understand it is attempting to reflect the analysis, but it’s the reverse association that is the likely “effect”, specifically, that organic foods are more healthful. It would be more approachable as something such as, Packaged foods labeled as organic have a more healthful profile than their conventional counterparts.

The conflicts of interest mention EWG. However, there is no mention about Novi Connect nor Amazon.com; both of those interests need to be addressed in the disclosure.

Please add specifics to the funding: "charitable foundation grants to EWG". Who are the foundations?

There is an extensive, extensive, list of references. Typically an original study has 20-40 references.

Introduction:

Lines 46-48: “As the availability of organic packaged and processed foods continues to grow [13], it is important to determine whether there are meaningful differences in the healthfulness of conventional and organic packaged and processed foods sufficient to justify choosing one  over  the  other.”
That argument excludes all the other reasons consumers may choose organic food, including sustainability reasons.

The authors state, quite forcefully, that two past studies that did not report differences in the nutritional “healthfulness” of conventional vs. organic packaged foods was because their methods were limited. Specifically, that a focus on nutritional profiling is too reductive. I disagree that examining nutrients and nutrient profiles is too reductive to compare the potential healthfulness of packaged foods. Importantly, the authors then in the next paragraph use nutritional composition to justify why assessing the processing level of foods is important for health. Also, the authors use the nutritional differences of their study to claim organic foods are “healthier”. That is not consistent. For an academic paper, the authors need to demonstrate that the level of processing of food is detrimental to  health, independent of nutritional composition, to maintain their earlier argument.

However, in the discussion at lines 345, the authors comment on other limitations of past studies. That could be noted in the introduction and then the introduction could be reduced down.

Methods

I, as with most readers, am not familiar with the EWG database. How representative of the US food purchases are the foods in this database? Are all of these foods, organic and conventional, available in the same supermarkets? Or could there be a bias in where these foods are offered?

The use of the NOVA classification is appropriate .

There is in sufficient detail on the food category classification and the “aisle” variable. The data are likely unbalanced so it would be helpful to see a summary of conventional and organic products within each food category.

Please add the statistical software used for analyses. If you are using SAS, there is a macro to compute the area under the ROC curve for predictive performance, which would benefit the analysis.

Results

Please analyze the data in Table 1 also with a mixed-effects model to consider the differences in food classifications. If you are using a t-test anyway for means, use a linear mixed regression model and compute the predicted values of each dependent variable from those models. Clarify the methods used to compute any p-values in table 1.

Table 2: The units for potassium are in grams/100 grams of food. Potassium is in mg and thus a one-unit (and thus a 1 gram) increment is not realistic. Translate that into a meaningful/appropriate difference for the odds ratio. Same issue for sodium. The odds ratios appear inflated.

Please add the units of measurement to Table 2 for # of ultra-processed ingredients.

The analysis would be greatly strengthened by a comparison to profile such as the NPI.

Discussion

There should be a discussion on the null findings for added sugar. Per table 1, all products had quite high levels of added sugar. The current text is simply restating the findings: “We did not observe relationships between organic status and added sugar or saturated fat content, two other nutrients associated with diet-related chronic disease. Results from our univariate and sensitivity analyses suggest that such relationships may exist. Future work should further study the possible relationships between added sugar and saturated fat in conventional and organic packaged foods.” Also, Table 1 did not adjust for food category. A discussion on the need to reduce added sugar content for both conventional and organic products is needed.

Overall, The discussion is basic and re-summarizes the findings and the introduction. It lacks a critical analysis. For example, there is no discussion on the magnitude of the effects. Odds ratios are reported, but what are the absolute differences based typical products purchased for example? Are they small, large? Will the health benefits add up with the more packaged foods purchased? How does the price difference for organic foods come into play? Should programs incentivize organic foods?

Reviewer 2 Report

Authors compared the content of organic and conventional packaged foods. The paper is well constructed and clearly written. I would like to congratulate the authors for the study. However, I have a reflection and I would like to know what the authors think about it. For me the content of organic products will always be better for health since regulations prohibit the use of many harmful ingredients in the production or preparation of organic products. For this reason, I do not fully understand what new information this study provides us?

Thank you. 

Round 2

Reviewer 1 Report

The revisions have improved the analysis and presentation.

The new table 3 (which is the previous table 1) still should be adjusted for food category via mixed-effects modeling. That approach will also account for the "imbalance" of organic vs. conventional foods, as was done for the mixed-effects logistic model.